# Unlocking Unlabeled Data: Ensemble Learning with the Hui-Walter Paradigm for Performance Estimation in Online and Static Settings

## Abstract

In the realm of machine learning and statistical modeling, practitioners often work under the assumption of accessible, static, labeled data for evaluation and training. However, this assumption often deviates from reality where data may be private, encrypted, difficult-to-measure, or unlabeled. In this paper, we bridge this gap by adapting the Hui-Walter paradigm, a method traditionally applied in epidemiology and medicine, to the field of machine learning. This approach enables us to estimate key performance metrics such as false positive rate, false negative rate, and priors in scenarios where no ground truth is available. We further extend this paradigm for handling online data, opening up new possibilities for dynamic data environments. Our methodology involves partitioning data into latent classes to simulate multiple data populations (if natural populations are unavailable) and independently training models to replicate multiple tests. By cross-tabulating binary outcomes across ensemble categorizers and multiple populations, we are able to estimate unknown parameters through Gibbs sampling, eliminating the need for ground-truth or labeled data. This paper showcases the potential of our methodology to transform machine learning practices by allowing for accurate model assessment under dynamic and uncertain data conditions.

## 1 Introduction

It is well known that data volume is growing rapidly. Most statistical, machine learning (ML), and deep learning techniques have developed when data were more scarce, and applications make increasingly unverified predictions based on data that is the same or similar to training data. Traditional statistical textbooks go to great lengths to emphasize the difference between interpolation and extrapolation, while warn of the dangers of the latter. Much of these data are delivered online (or streamed) and unlabeled. Researchers attempting to move machine learning online still assume that the data are mostly labeled (Gomes et al., 2019). Additionally, performance evaluation is a necessary and often daunting task when using a ML system in the production environment. Performance is often difficult to measure due to insufficient quantities and the variety of ground truth data labeled. Furthermore, data can be confidential, encrypted, or impossible (including costly) to measure, and the data landscape for online or streaming algorithms can constantly change. The actual distribution of the population is often unknown. Thus, it is impossible to relatively easily compare the output distribution of ML systems with the distribution of the population. Fortunately, the problems described are not limited to machine learning.

Epidemiology is another discipline that relies on accurate efficacy measurements with insufficient ground truth and population data. Disease testing is the most recognizable example of epidemiology. Epidemiologists distinguish between efficacy and effectiveness when talking about moving from a laboratory environment where controlled experiments are possible (efficacy), whereby the test is deployed to test-recipient populations (effectiveness) (Singal et al., 2014). By analogy, a laboratory-developed Covid test measures efficacy by false-negative and false-positive rates in a double-blind controlled trial. The effectiveness is then measured by deploying the test to a remote population. In an applied data science environment, this mimics the applied

data scientist who has trained a model in a notebook with labeled data and then deployed the model to production. Thus, the method we propose borrows concepts from how epidemiologists measure efficacy or effectiveness and apply them to measuring the performance of ML systems in production environments where there is no gold standard or ground truth. Our approach can be applied to any field in which unsupervised learning is ubiquitous and is generally likely to be of interest to anyone working in an applied data science environment. Alternatives to our approach are not common. Based on our research into this area, the most similar approach is analyzing agreement between multiple classifiers to arrive at an estimate of performance; we provide comparisons of our approach to this strategy.

We use the Hui-Walter paradigm, which uses Bayesian methods to estimate the prior probability (prevalence) and false positive and false negative rates in ensemble models trained to detect the results of running in production or online (Hui & Walter, 1980). This estimate uses binary results of diagnostic tests, i.e. Covid positive or negative, for multiple tests and multiple populations in a table. We then performed Gibbs sampling to estimate unknown parameters, priors, false positive, and false negative rates without relying on ground truth data. We can make these estimates because more than one population provides us with enough equations and unknowns to solve the parameters of interest. We expand this idea by introducing a closed form solution in which two populations naturally occur, i.e. multiple data centers or regions, and use it to extend this framework to the online environment. When only one population is available, we use latent structure models for both online and offline methods. Many of the applications use bagging techniques or other ensemble methods in many settings. These ensemble methods naturally extend to our methodology. In our experiments, we divided the Wisconsin breast cancer and Adult data sets into latent classes for demonstration purposes, to simulate scenarios with multiple population datasets, and train independent models for simulation of multiple tests.

## 2 Proposed Method

The Hui-Walter paradigm is a technique epidemiologists use to measure effectiveness without establishing the fundamental truth. The paradigm is based on the comparison of multiple tests and their results applied to more than one logical subset of the population (Hui & Walter, 1980). When evaluating ML systems, tests are analogous to binary categorizers, results are predictions (i.e. a 1 or 0 binary classification corresponds to a positive or negative test result), and population subsets are groups of observations that are considered to be in distinct subpopulations. To obtain different subpopulations when only one exists, which may be the case when evaluating some ML systems, we propose using latent structural analysis, consisting of latent class analysis (LCA) or latent profile analysis (LPA), to split the data into two populations. Of course, some ML systems may already use multiple distinct subpopulations; for example, a practitioner has different data centers or databases to choose from. The framework we propose is two-fold. For machine learning practitioners working in an industry where multiple populations are often available for selection, we may consider the Hui-Walter paradigm applied to two or more models trained to detect results based on these data. If only one population is present, we use LPA or LCA to separate our data into multiple latent populations. Next, we give a closed form solution applying Hui-Walter to the data stream, measuring prior probabilities and false positive and false negative rates for unlabeled data streams. Our proposed solution dynamically assigns latent classes to streaming data and calculates unknown performance metrics for unlabeled data streams.

### 2.1 Review

#### 2.1.1 Latent Class and Latent Profile Analysis

*Latent structure analysis* is a statistical method used to identify patterns or structures that underlie data. Latent class analysis is commonly used in psychology, sociology, and public health to identify subgroups or classes in a population based on their responses to a set of categorical variables. Similarly, researchers use latent profile analysis for continuous variables, and both models are generalized under the latent structure analysis moniker. They are also valuable tools for data scientists interested in understanding the underlying structure of a population and identifying subpopulations based on categorical, ordinal, or continuous variables Goodman (1974). In the analysis of latent profiles, we assume that each latent class follows a normal distribution of features. Note that the normality assumption only applies to these subpopulations and not to

their combined distribution, which we model as a Gaussian mixture model. Each latent $(1, \ldots, K)$ class has a density component in the mixed model.

### 2.1.2 Maximum Likelihood Estimate for Product Multinomial Distributions

Maximum likelihood estimation (MLE) is a method for estimating the parameters of a statistical model given a set of observations. One common application of MLE is to estimate the parameters of the multinomial distribution (Poleto et al., 2014).

The multinomial distribution is a generalization of the binomial distribution that allows for more than two possible outcomes. Given a set of $N$ independent trials, the multinomial distribution describes the probability of observing a specific combination of counts. The probability mass function of the multinomial distribution is given by $P(x_1, x_2, \ldots, x_k | n, p_1, p_2, \ldots, p_k) = \frac{n!}{x_1! x_2! \ldots x_k!} p_1^{x_1} p_2^{x_2} \ldots p_k^{x_k}$, where $x_i$ is the number of times the $i$-th outcome was observed, $n$ is the total number of trials, and $p_i$ is the probability of the $i$-th outcome occurring.

To perform MLE on the multinomial distribution, we wish to find the values of $p_1, p_2, \ldots, p_k$ that maximize the likelihood of the observed data. We can do this by taking the product of the probabilities of each outcome and maximizing this product for the parameters. This procedure is equivalent to maximizing the log-likelihood function $\mathcal{L}(\mathbf{p}) = \sum_{i=1}^{k} x_i \log p_i$. The log-likelihood function is easier to work with because the product of many small numbers can be unstable, while the sum of the logs is more stable for calculation.

We can find the maximum likelihood estimates by setting the partial derivatives of the log-likelihood function to zero and solving for $p_i$. This reasoning leads to the following unbiased estimators $\hat{p}_i = \frac{x_i}{n}$.

The *product-multinomial* distribution is the distribution of multiple independent multinomial distributions. Extending the MLE to product-multinomial distributed data is as follows for $n$ independent multinomial distributions:

The likelihood function is given by $L(\boldsymbol{X}) = \prod_{i=1}^{n} \prod_{j=1}^{k} p_{ij}^{x_{ij}}$ where $n$ is the number of populations, $k$ is the number of categories, $x_{ij}$ is the number of times the category $j$ was observed in the population $i$ and $\boldsymbol{X}$ is a vector of the parameters of the model.

The maximum likelihood estimate of the parameters, denoted by $\hat{\boldsymbol{X}}$, is found by maximizing the likelihood function of the parameters:

$$\hat{\boldsymbol{X}} = \arg\max_{\boldsymbol{X}} L(\boldsymbol{X})$$

We can estimate this using numerical optimization techniques.

The maximum likelihood estimate is not necessarily the best estimate of the parameters, and there are other methods of estimation, such as the method of moments and Bayesian estimation, that may be more appropriate in some cases. However, the maximum likelihood estimation is a widely used and well established method with several attractive properties, such as asymptotic efficiency (meaning that it becomes more accurate as the number of observations increases)(Hui & Walter, 1980).

In general, MLE is a valuable method for estimating the parameters of the product-multinomial distribution and has attractive theoretical properties. However, it is crucial to consider the method's limitations, especially when working with small sample sizes.

### 2.1.3 Gibbs Sampling

Gibbs sampling, named after the physicist Josiah Willard Gibbs, is an analogy between the sampling algorithm and statistical physics. The algorithm was described by brothers Stuart and Donald Geman in 1984, some eight decades after the death of Gibbs, in their work on algorithmic image restoration (Geman & Geman, 1984). Gibbs sampling is a Markov chain Monte Carlo (MCMC) technique, typically the Metropolis-Hastings algorithm, for estimating the marginal distribution of variables in a multivariate distribution.

In Gibbs sampling, we start with an initial guess for the values of the variables and then iteratively sample each variable from its conditional distribution given the current values of the other variables. This process is repeated many times to obtain a sample from the joint distribution of the variables. Then we use the sample to estimate the marginal distribution of each variable or to compute other statistical quantities of interest.

Gibbs sampling has several advantages over other MCMC techniques. It is relatively simple to implement and easy to obtain good mixing (i.e., the convergence of the samples to the target distribution) in practice. It is also well suited to parallelization, which can speed up the sampling process.

However, Gibbs sampling can be inefficient when the variables are highly correlated or anticorrelated, as it can take many iterations to explore the full joint distribution. It is also sensitive to the choice of initial values and can get stuck in local modes for poorly chosen initial values.

Gibbs sampling finds applications in statistical physics, machine learning, and other fields where it is necessary to sample from multivariate distributions. It is beneficial for estimating the posterior distribution in Bayesian statistical models and is usually used with other MCMC techniques such as Metropolis-Hastings sampling.

Given a multivariate distribution, Gibbs sampling allows us to sample from a conditional distribution rather than to marginalize by integrating over a joint distribution; the joint distribution is often not known explicitly or difficult to sample from directly (Johnson et al., 2001). For purposes of our proposed approach in the next section, Gibbs sampling allows us to assume prior uncertainty for the prevalence (i.e. base rate) and classifier performance parameters (false positive rates, false negative rates) and represent them as independent distributions.

The algorithm is shown in Algorithm 1:

---
**Algorithm 1** Gibbs Sampling

---
1: **procedure** GIBBSSAMPLE($x_1, \ldots, x_n, T$)
2:      Initialize $X^{(0)} = (x_1, \ldots, x_n)$
3:      **for** $t = 1, 2, \ldots, T$ **do**
4:          **for** $i = 1, \ldots, n$ **do**
5:              Sample $X_i^{(t)}$ from $p(X_i^{(t)} \mid X_1^{(t)}, \ldots, X_{i-1}^{(t)}, X_{i+1}^{(t-1)}, \ldots, X_n^{(t-1)})$
6:          **end for**
7:      **end for**
8:      **return** samples $X^{(0)}, \ldots, X^{(T)}$
9: **end procedure**

---

This algorithm takes a set of variables $X = (x_1, \ldots, x_n)$ and $T$ as input and returns $T$ samples from their joint distribution. It does this by iteratively sampling each variable from its conditional distribution given the current values of the other variables. The algorithm ends after a fixed number of iterations $T$.

### 2.2 Proposed

### 2.2.1 Hui-Walter

The Hui-Walter paradigm is a framework from epidemiology for estimating the false positive rate, false negative rate, and prior probabilities across multiple populations, with more than one test. In the epidemiological context, the Hui-Walter paradigm was designed for disease tests, i.e. Covid tests, where the results are 0 or 1 for does not-have-the-disease and has-the-disease, respectively. Since we are applying this paradigm to the machine learning context, we will discuss the method in terms of classifications from binary categorizers. The Hui-Walter paradigm suggests that we can form a $n \times m \times k$ contingency table for $n$ binary categorizers, $m$ populations, and $k$ classes. The goal of this method is to estimate a parameter vector $(\hat{\alpha}, \hat{\beta}, \hat{\theta})$ for each model and population, where $\alpha$ is the false positive rate or $1 -$ specificity, $\beta$ is the false negative rate or $1 -$ sensitivity, and $\theta$ is the prior probability of the positive class in the population. The Hui-Walter paradigm is valid for $m$ populations, $n$ classes, and $k$ categorizers where $m, n, k \in \mathbb{N}$. When $m = n = k = 2$, we can show this result using maximum likelihood estimation Hui & Walter (1980).

Figure 1: Two tests on one population.

For one population, consider Figure 1. One population is multinomial distributed cell data. The cells $X_1, X_4$ are where both models ($T_1$ and $T_2$) agree, and $X_2, X_3$ are where the models disagree.

Now, the advantage of maximum likelihood for a single population is that $p_1$ is the probability that both tests produce a positive result. We solve this over-determined system by minimizing the likelihood function. In our simplified setup, we have a parameter vector $p = (p_1, p_2, p_3, p_4)$, which we wish to estimate. Because these probabilities in question are the outcomes of binary classifiers or diagnostic tests, we may express them in terms of the false positive and false negative rates of the models in question and the population's prior probability. Let $\alpha_1, \alpha_2$ be the false positive rates for $T_1$ and $T_2$, let $\beta_1, \beta_1$ be the false negative rates for $T_1$ and $T_2$, and let $\theta$ be the prior probability for the positive class (base rate) of the population (or the disease in the epidemiological case).

$$
\begin{aligned}
p_1 &= \theta(1 - \beta_1)(1 - \beta_2) + (1 - \theta)\alpha_1\alpha_2 \\
p_2 &= \theta(1 - \beta_1)\beta_2 + (1 - \theta)\alpha_1(1 - \alpha_2) \\
p_3 &= \theta\beta_1(1 - \beta_2) + (1 - \theta)(1 - \alpha_1)\alpha_2 \\
p_4 &= \theta\beta_1\beta_2 + (1 - \theta)(1 - \alpha_1)(1 - \alpha_2)
\end{aligned}
\tag{1}
$$

We need to consider two populations because the base rates of a single population are unknown. We assume that $\alpha_i$ and $\beta_i$ are the same in both populations for $i = 1, 2$, therefore we have six equations with six unknowns. In other words, $p_i$ where $i = 1, 2, \ldots 8$, can be written algebraically as combinations of 6 variables. Therefore, we use two populations so that the above estimates are solvable (Johnson et al., 2001). At least two populations guarantee that we have as many equations as unknowns. For two populations, we have two independent multinomial distributions. Therefore, the likelihood function is the product of the likelihood functions (Hui & Walter, 1980). Now consider two populations, $P_1$ and $P_2$ with prior probabilities $\theta_1$ and $\theta_2$. In that case, our two-by-two-cell data becomes Figure 2.

Now that we have two populations, we can solve the prior probabilities to eliminate the unknowns in our estimates. Because we have two independent multinomial distributions, which can be considered a product-multinomial distribution, the likelihood function is the product of the individual likelihood functions adjusted with the sample size. Let $n_1, n_2$ be the total sample size of each population, then the likelihood function is

$$
l = l_1 l_2 = \prod_{j}^{2} \prod_{i}^{2} p_{i,j}^{x_{ij}}
$$

Where $p_{i,j}$ is the probability of observing the $j$ th outcome in the $i$ th population.

We want to estimate the parameter vector $\boldsymbol{X} = (\alpha, \beta, \theta)$. A real solution with coordinates that satisfy these constraints is a maximum likelihood estimate.

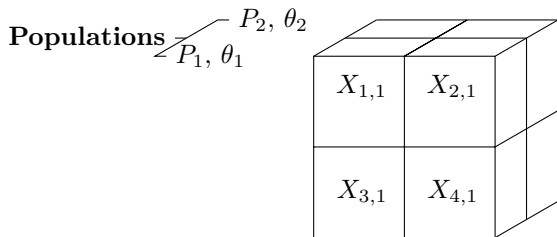

Figure 2: Hui-Walter assumptions with a $2 \times 2 \times 2$ contingency table. Variables $X_{1,1}, X_{2,1}, X_{3,1}$ and $X_{4,1}$ are cell frequency counts from a product-multinomial distribution. This setup is the minimum number of populations and classifiers required for the Hui-Walter method. However, this framework supports $n$ populations and $m$ classifiers.

For our simplified case with two populations we get $\frac{\partial L}{\partial p_i} = \frac{x_i}{p_i} - \lambda = 0, \frac{\partial L}{\partial p_j} = \frac{x_j}{p_j} - \mu = 0$ for $i = 1, \ldots, 4$, and for $j = 5, \ldots, 8$. The maximum likelihood then is $\mathbf{p} = (\frac{p_1}{n_1}, \ldots, \frac{p_4}{n_1}, \frac{p_5}{n_2}, \ldots, \frac{p_8}{n_2})$ from 1. This closed formula gives the maximum likelihood estimate for the product-multinomial distribution. This MLE extends naturally to setups with more than 2 classes.

In other cases, when the points of the global maximum are complex or lie outside the unit hypercube, estimates can be obtained by numerical maximization of $l$ with the solution restricted to be within the unit hypercube, again subject to appropriate constraints. Asymptotically, because the observed proportions of cell counts are continuous functions of the parameters, the maximum likelihood solution will converge to $(\alpha, \beta, \theta)$. Let $L = ln(l)$, then we can formulate the information matrix

$$\mathcal{I}(\boldsymbol{X}) = \mathbb{E}[\mathbf{Hess}(\mathbf{X} \otimes \mathbf{X})]$$

In other words, when the MLE resides within the interior point of the feasible set, we can solve for the information matrix by taking the expectation value of the hessian of the outer product of our parameter vector, $\boldsymbol{X}$ (Blasques et al., 2018).

Inverting this matrix gives the variance and covariance matrix. Now that we have shown that there is a maximum likelihood estimate for the parameter vector $(\alpha, \beta, \theta)$ exists for the $2 \times 2 \times 2$ case, it is trivial to see that this generalizes to the $n \times m \times k$ case for arbitrary dimensions of the population outcome tensor. We can generally find a solution to the parameter vector using Bayesian methods such as Monte Carlo estimates or Gibbs sampling to estimate our parameter vector, $\boldsymbol{X} = (\alpha, \beta, \theta)$.

One of the assumptions of Hui-Walter is that the tests (or ML binary classifiers) be conditionally independent Hui & Walter (1980). We can test for conditional independence, which can easily be tested by performing a goodness-of-fit hypothesis test on the contingency table with a $G$-test (Hui & Zhou, 1998).

### 2.2.2 Hui-Walter Online

Online Machine Learning refers to training machine learning models on data that arrive continuously and in real time rather than using a static data set. In this setting, the model updates its parameters as new data arrive and makes predictions accordingly (Gomes et al., 2019). This approach is helpful in applications such as streaming data analysis, recommendation systems, and dynamic pricing.

One of the key benefits of online machine learning is its ability to handle large amounts of data efficiently, which is especially important in scenarios where data volume is increasing rapidly. Additionally, online machine learning algorithms can adapt to changing data distributions, which is critical in applications where the underlying data distribution constantly evolves. This process enables the models to provide accurate predictions even in dynamic environments, improving the overall system performance.

Various algorithms work online, including stochastic gradient descent, k-means, and online support vector machines (Gomes et al., 2019). These algorithms differ in their update rules, but all aim to minimize the cumulative loss incurred over time as new data arrive. A critical aspect of online machine learning is the

choice of the loss function, which determines the trade-off between the accuracy of the model and its ability to adapt to new data. However, these methods are still limited to using labeled data and assuming that ground truth is available. Online machine learning provides a powerful and flexible approach to real-time training machine learning models, but fails in some applied settings where the data is encrypted.

We saw previously that the maximum likelihood estimate for the product-multinomial distribution is $\mathbf{p} = (\frac{p_1}{n_1}, \ldots, \frac{p_4}{n_1}, \frac{p_5}{n_2}, \ldots, \frac{p_8}{n_2})$. We use this estimate to come up with a naive estimate of the base rate for a population taking $\frac{X_i}{n}$ for $i = 1, 2$. We use this estimate to solve for the other parameters related to false-positive and false-negative rates and keep a tally of the streaming data as a time series. We then use the prior probability of our base class from our training environment to test for significant prior drift.

A closed-form solution to the Hui-Walter estimates without a confidence interval appears in (Hui & Walter, 1980; Enùe et al.). We reformulate our contingency table as follows:

|  | $T_2$ (pos) | $T_2$ (neg) |  |
|---|---|---|---|
| $T_1$ (pos) | $a_i$ | $b_i$ | $g_i$ |
| $T_1$ (neg) | $c_i$ | $d_i$ | $h_i$ |
|  | $e_i$ | $f_i$ | $n_i$ |

where $i$ ranges over each population. For simplicity, we will focus on $i = 2$ but mention that this can be extended to more populations and classes. By computing the row sums and column sums of this matrix, we look at the following discriminant:

$$F = \pm\sqrt{\left(\frac{g_1 e_2 - g_2 e_1}{n_1 n_2} + \frac{a_1}{n_1} - \frac{a_2}{n_2}\right)^2 - 4\left(\frac{g_1}{n_1} - \frac{g_2}{n_2}\right)\frac{a_1 e_2 - a_2 e_1}{n_1 n_2}} \tag{2}$$

If this discriminant is zero or complex, then this online method cannot be calculated (Hui & Walter, 1980). One possible explanation for this phenomenon is the existence of Simpson's paradox and the algebraic geometry that arises when dealing with three-way contingency tables (Slavkovíc et al., 2009). The closed solutions to the Hui-Walter estimates are as follows:

$$\hat{\theta}_1 = \frac{1}{2} - \left[\frac{g_1}{n_1}\left(\frac{e_1}{n_1} - \frac{e_2}{n_2}\right) + \frac{g_1}{n_1}\left(\frac{g_1}{n_1} - \frac{g_2}{n_2}\right) + \frac{a_2}{n_2} - \frac{a_1}{n_1}\right]\frac{1}{2F}$$

$$\hat{\theta}_2 = \frac{1}{2} - \left[\frac{g_2}{n_2}\left(\frac{e_1}{n_1} - \frac{e_2}{n_2}\right) + \frac{g_2}{n_2}\left(\frac{g_1}{n_1} - \frac{g_2}{n_2}\right) + \frac{a_2}{n_2} - \frac{a_1}{n_1}\right]\frac{1}{2F}$$

$$\hat{\alpha}_1 = \left(\frac{g_1 e_2 - e_1 g_2}{n_1 n_2} + \frac{a_2}{n_2} - \frac{a_1}{n_1} + F\right)\left[2\left(\frac{e_2}{n_2} - \frac{e_1}{n_1}\right)\right]^{-1}$$

$$\hat{\alpha}_2 = \left(\frac{g_2 e_1 - e_2 g_1}{n_1 n_2} + \frac{a_2}{n_2} - \frac{a_1}{n_1} + F\right)\left[2\left(\frac{g_2}{n_2} - \frac{g_1}{n_1}\right)\right]^{-1} \tag{3}$$

$$\hat{\beta}_1 = \left(\frac{f_1 h_2 - h_1 f_2}{n_1 n_2} + \frac{d_2}{n_2} - \frac{d_1}{n_1} + F\right)\left[2\left(\frac{e_2}{n_2} - \frac{e_1}{n_1}\right)\right]^{-1}$$

$$\hat{\beta}_2 = \left(\frac{f_2 h_1 - h_2 f_1}{n_1 n_2} + \frac{d_2}{n_2} - \frac{d_1}{n_1} + F\right)\left[2\left(\frac{g_2}{n_2} - \frac{g_1}{n_1}\right)\right]^{-1}$$

Using the formulas mentioned above, we can increase the streams' counts and apply the estimates to the streaming data. This allows us to investigate the false positive and false negative rates, as well as the prior probabilities of streaming data at a specific time step. Interestingly, boosting provides an approach to this problem. It is a family of algorithms that convert multiple weak learners working in parallel into strong learners. This is where Hui-Walter online comes into play. With it, we can harness the power of these weak learners to assess the false-positive and false-negative rates, enhancing predictions by making them "prior

aware" of the unlabeled data streams. Such an approach is not unique. In fact, several learning frameworks, such as the one mentioned by (Davis, 2020), advocate for better predictions by making the learner aware of previous information. However, a point to be taken into account is the sign of $F$ from equation 2. As pointed out by (Hui & Walter, 1980; Johnson et al., 2001), it can be either positive or negative, and sometimes, it does not present a solution at all. Hence, its value is determined by what yields *plausible* solutions, specifically within a probability simplex.

Our online algorithm runs as follows in Algorithm 2:

---
**Algorithm 2** Online Hui-Walter on Streaming Data
---
1: **procedure** ONLINE HUI-WALTER($\mathcal{X}, T$)
2:      $t \leftarrow 1$
3:      Initialize models $f_1, f_2$ with $\mathcal{X}$
4:      Initialize three-way contingency table $\mathcal{T}$ and history $\mathcal{H}$
5:      **while** $t \leq T$ **do**
6:          Receive instance $x_t$
7:          Predict labels $y_1 = f_1(x_t), y_2 = f_2(x_t)$
8:          Add $x_t$ to $\mathcal{H}$
9:          Get LPA profiles of $\mathcal{H}$
10:         Update cell counts of $\mathcal{T}$
11:         **if** t » n **then**
12:            Calculate $\mathbf{X} = (\theta, \alpha, \beta))$
13:         **end if**
14:         $t \leftarrow t + 1$
15:      **end while**
16: **end procedure**
---

In Algorithm 2, $\mathcal{X}$ is the data set and $f_1$ and $f_2$ trained on subsets of the columns, $T$ is the number of instances to receive, $\mathcal{T}$ is a contingency table with as many dimensions as populations, $\mathcal{H}$ is the history of previously seen samples. The implementation of $\mathcal{H}$ could cache the previously seen samples in an LRU cache. The procedure calculates the latent profiles on the history $\mathcal{H}$ to assign the data to a population. After the $n$ instances have been predicted, the estimates for $\theta, \alpha, \beta$ start to be calculated and updated. The selection of $n$ should be reasonable such that there is enough stability in the values for $\theta, \alpha, \beta$ (lessening variation from one instance to the next) without sacrificing too many of the samples. This analogous to a "burn-in" concept commonly seen in MCMC (Hamra et al., 2013), but keep in mind that we are not discarding samples or assuming stationarity has been achieved.

## 3 Experimental Results

### 3.1 Data Sets

#### 3.1.1 Wisconsin Breast Cancer

The Wisconsin breast cancer data set is a clinical data collection used to analyze breast cancer tumors. It was created in the early 1990s by Dr. William H. Wolberg, a physician and researcher at the University of Wisconsin-Madison. The data set contains 569 observations and includes information on patient characteristics, such as age, menopausal status, and tumor size, as well as diagnosis and treatment of cancer. In the diagnosis of 569 observations, 357 were benign and 212 were malicious.

The Wisconsin Breast Cancer Data Set has been widely used in machine learning and data mining to predict breast cancer diagnosis and prognosis. Researchers have used the data set to develop algorithms and models to classify tumors as benign or malignant and predict the probability of breast cancer recurrence or survival. These predictions can guide the treatment of patients with breast cancer and identify potential risk factors for the disease. However, it is essential to note that the data set has some limitations. For example, data is based on a specific population of patients and may not represent all cases of breast cancer. Furthermore, the

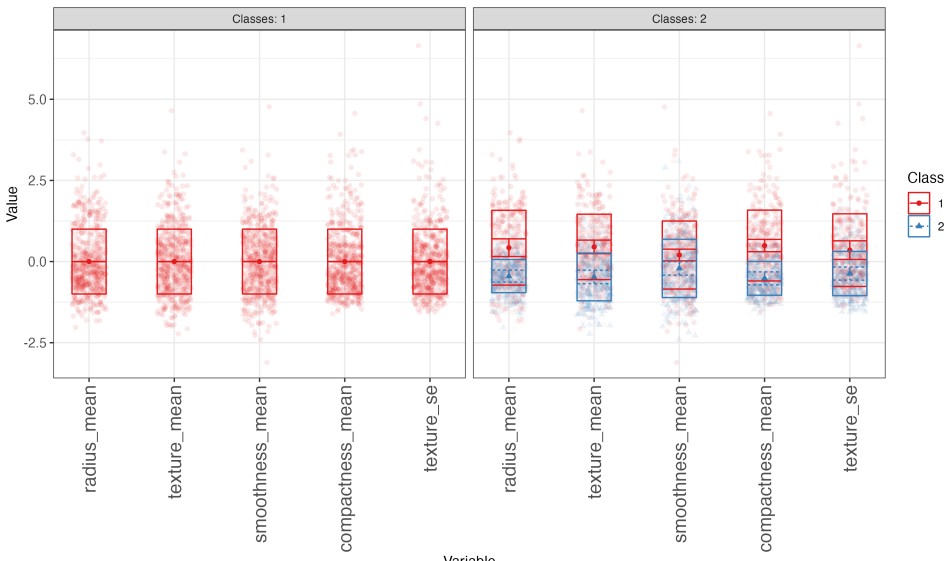

Figure 3: Five features from the Wisconsin Breast Cancer Data Set examined with Latent Profile Analysis

data are from the 1990s and may not reflect more recent advances in breast cancer diagnosis and treatment. Despite these limitations, the Wisconsin breast cancer data set remains an essential resource for researchers studying breast cancer and working to improve the diagnosis and treatment of this disease. It has contributed significantly to our understanding of breast cancer and will continue to be a valuable resource for researchers in the field.

In addition to its role in medicine, it has been used as an almost canonical data set for data science education and benchmarking of ML algorithms. It is one of the most downloaded data sets on the UCI Machine Learning Repository(Dua & Graff, 2017)[1].

### 3.1.2 Adult

This is another canonical data set from the UCI Machine Learning Repository(Dua & Graff, 2017) [2]. It contains census data from 1994 for the task of predicting whether an adult's income exceeds $50k/year. There are 32,561 observations described with a mix of categorical and continuous characteristics such as age, sex, education level, occupation, capital gains, and capital losses.

### 3.2 Hui-Walter Data Experiments

The core findings of our experimental results are that for each experimental data set in the static context, we found the true parameters of interest inside the confidence intervals produced by Gibbs Sampling.

### 3.2.1 Wisconsin Breast Cancer Data Set

**Latent Profile Analysis** We performed a Latent Profile Analysis (LPA) on the Wisconsin Breast Cancer data set to obtain two populations. Due to collinearities in the data set, a subset of features was selected to estimate the latent profiles. This subset was determined via hierarchical clustering on Spearman rank-order correlations and selecting one feature from each cluster, where clusters were defined as being separated by at least a distance of 1 per Ward's linkage. LPA, assuming varying variances and varying covariances, was used to fit models assuming 1 through 10 latent classes. The approximate weight of evidence (AWE) criterion was used to identify the optimal number of classes, which was 2. AWE was used instead of BIC for model

---

[1]https://archive.ics.uci.edu/ml/datasets/Breast+Cancer+Wisconsin+%28Diagnostic%29
[2]https://archive.ics.uci.edu/ml/datasets/Adult

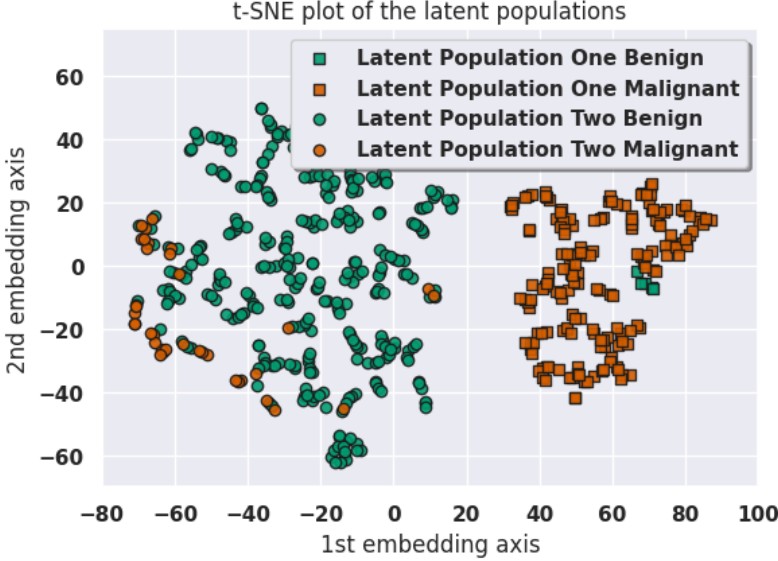

Figure 4: T-SNE dimension reduction of both latent populations.

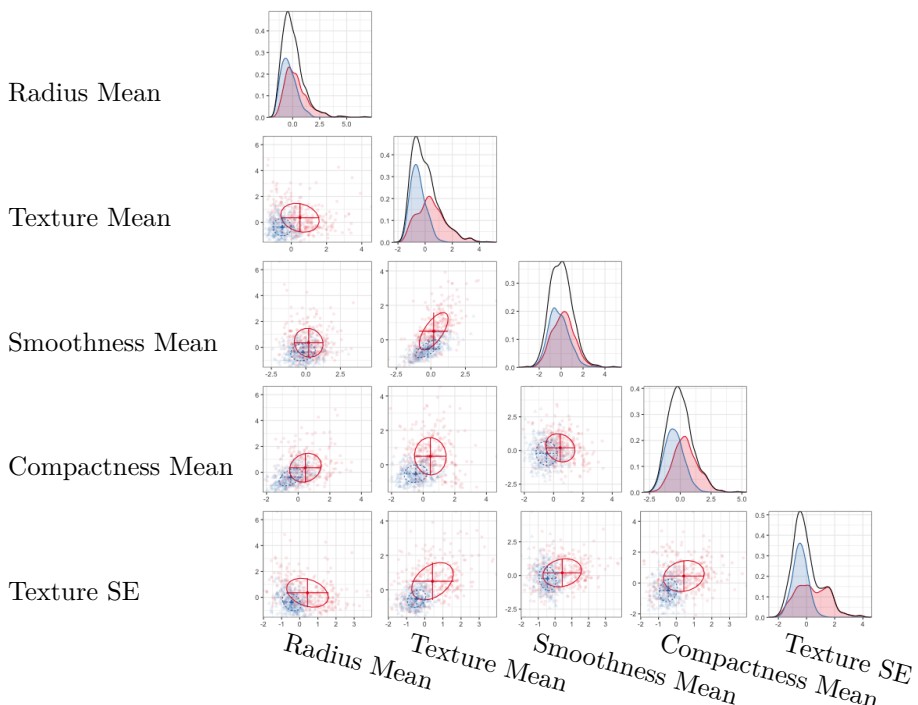

Figure 5: Bivariate plot of two Latent Profiles.

selection due to its ability to select more parsimonious models (Banfield & Raftery, 1993), especially relevant because we assumed complex parameterization (varying variances and varying covariances).

In Figure 5, we can see two distinct latent classes or subpopulations among the data. Figure 3 shows the columns of the data partitioned into one and then two latent classes. In Figure 4, we performed the t-SNE (Maaten & Hinton, 2008) dimension reduction of the columns of the Wisconsin Breast Cancer data set and labeled it according to the latent classes found with the analysis of the latent profile. We see that latent profile analysis and t-SNE separated the data into two latent populations.

**Results**  We trained a random forest model (RF) and a support vector classifier (SVC) model using the data after the two latent populations have been assigned. To ensure that the models are independent, we chose mutually exclusive columns of the data for each model. Then, we split the data set into a training and test data set, sizes 200 and 369, respectively. Finally, we trained the RF and SVC models on the training data set.

Table 1: Model Predictions Over The Two Latent Populations

|  |  | Model Outcome | |
| --- | --- | --- | --- |
| Population | Model | Malignant | Benign |
| Population One | RF | 109 | 80 |
|  | SVC | 89 | 100 |
| Population Two | RF | 10 | 170 |
|  | SVC | 19 | 161 |

Table 1 contains a three-way contingency table of categorizations on the Wisconsin Breast Cancer data set within each latent population based on the RF and SVC models. Both models predicted different proportions of malignant to benign tumors for the populations, suggesting models have some degree of independence.

With the models trained, we then proceeded to estimate parameters $\hat{\alpha}$, $\hat{\beta}$, and $\hat{\theta}$ without the ground truth labels by using the Hui-Walter framework with Gibbs Sampling. The sensitivity and specificity were each instantiated with a $Beta(1,1)$ distribution.

Table 2: Wisconsin Breast Cancer Data

| Experimental Results | | | | |
| --- | --- | --- | --- | --- |
| Model | TPR | TNR | FPR ($\hat{\alpha}$) | FNR ($\hat{\beta}$) |
| RF | 0.690 | 0.956 | 0.044 | 0.310 |
| SVC | 0.547 | 0.899 | 0.101 | 0.453 |
| Population | Prior ($\hat{\theta}$) | | | |
| Population One | 0.819 | | | |
| Population Two | 0.035 | | | |

| True Values | | | | |
| --- | --- | --- | --- | --- |
| Model | TPR | TNR | FPR | FNR |
| RF | 0.789 | 0.969 | 0.031 | 0.211 |
| SVC | 0.549 | 0.868 | 0.132 | 0.451 |
| Population | Prior | | | |
| Population One | 0.635 | | | |
| Population Two | 0.122 | | | |

The estimated parameters using Hui-Walter with Gibbs Sampling are shown along with the actual values based on the ground truth in Table 2. For the SVC model, Hui-Walter estimated $\hat{\alpha}$ within 0.2 (0.101 compared to 0.132) and $\hat{\beta}$ within 0.01 (0.453 compared to 0.451). For the RF model, Hui-Walter estimates for $\hat{\alpha}$ was within 0.02 (0.044 compared to 0.031) and $\hat{\beta}$ was within 0.1 (0.310 compared to 0.211). The estimated priors $\hat{\theta}$ for Populations One and Two were close as well; Hui-Walter overestimated the prevalence in Population One (0.819 compared to 0.635) and underestimated for Population Two (0.035 compared to 0.122).

In Table 3, we have the confidence intervals for each parameter estimate. All the confidence intervals for the errors contain the parameter we found during the evaluation phase of the experiment. However, the true

prevalence (prior or base rate) is outside the confidence interval for both populations ($\hat{\theta}_1$ and $\hat{\theta}_2$). Depending on context, these margins of error may be acceptable as a measure of classifier performance without requiring ground truth labeled data to compute.

Table 3: Estimated Parameters

| Parameter | CI | Mean | SD |
|---|---|---|---|
| $\hat{\theta}_1$ | (0.693, 0.947) | 0.819 | 0.064 |
| $\hat{\theta}_2$ | (3.9e-07, 0.084), | 0.035 | 0.026 |
| $\hat{\alpha}_1$ | (0.004, 0.081) | 0.044 | 0.020 |
| $\hat{\alpha}_2$ | (0.054, 0.150) | 0.101 | 0.025 |
| $\hat{\beta}_1$ | (0.198, 0.425) | 0.310 | 0.058 |
| $\hat{\beta}_2$ | (0.357, 0.547) | 0.453 | 0.049 |

### 3.2.2 Adult

In lieu of latent profile analysis, we used the sex of the adult to partition the data set, resulting in two populations of $10,771$ females and $21,790$ males. We performed an $\frac{80}{20}$ split on the Adult data set to produce a training and test data set, sizes $26,048$ and $6,513$, respectively. We trained two classifiers, a linear regression (LR) and a random forest classifier (RF), on different columns of the data in the training data set.

Table 4: Model Predictions Over The Two Subpopulations

| Population | Model | Model Outcome | |
|---|---|---|---|
| | | $50K & Under | Over $50K |
| Female | LR | 2003 | 123 |
| | RF | 1968 | 158 |
| Male | LR | 3521 | 866 |
| | RF | 3258 | 1129 |

Table 4 contains a three-way contingency table of categorizations on the Adult data set within each population based on the LR and RF models. Both models predicted similar proportions of income for the populations.

With the models trained, we then proceeded to estimate parameters $\hat{\alpha}$, $\hat{\beta}$, and $\hat{\theta}$ without the ground truth labels by using the Hui-Walter framework with Gibbs Sampling. The sensitivity and specificity were each instantiated with a $Beta(1,1)$ distribution.

Table 5: Adult Data Set

| Experimental Results | | | | |
|---|---|---|---|---|
| Model | TPR | TNR | FPR ($\hat{\alpha}$) | FNR ($\hat{\beta}$) |
| LR | 0.608 | 0.992 | 0.008 | 0.392 |
| RF | 0.793 | 0.990 | 0.010 | 0.207 |
| Population | Prior ($\hat{\theta}$) | | | |
| Female | 0.084 | | | |
| Male | 0.316 | | | |

| True Values | | | | |
|---|---|---|---|---|
| Model | TPR | TNR | FPR | FNR |
| LR | 0.446 | 0.942 | 0.058 | 0.554 |
| RF | 0.523 | 0.906 | 0.094 | 0.477 |
| Population | Prior | | | |
| Female | 0.110 | | | |
| Male | 0.305 | | | |

Table 6: Estimated Parameters

| Parameter | CI | Mean | SD |
|---|---|---|---|
| $\hat{\theta}_1$ | (0.0656, 0.103) | 0.084 | 0.010 |
| $\hat{\theta}_2$ | (0.293, 0.338), | 0.316 | 0.011 |
| $\hat{\alpha}_1$ | (1.09e-4, 0.015) | 0.008 | 0.004 |
| $\hat{\alpha}_2$ | (5.15e-6, 0.020) | 0.010 | 0.006 |
| $\hat{\beta}_1$ | (0.353, 0.428) | 0.392 | 0.019 |
| $\hat{\beta}_2$ | (0.165, 0.247) | 0.207 | 0.021 |

The Hui-Walter method obtained close estimates of some of the parameters. The 95% confidence intervals for the rates of adults with over \$50$K$ income contained the ground truth rate for the male population ($\theta_2$) and was 0.007 within the ground truth rate of the female population ($\theta_1$). The confidence intervals for the error rate estimates ($\hat{\alpha}_1$, $\hat{\alpha}_2$, $\hat{\beta}_1$, $\hat{\beta}_2$) did not contain the actual values. However, $\hat{\alpha}_1$ and $\hat{\alpha}_2$ underestimated the actual by 7-10 times (0.008 compared to 0.058; 0.010 compared to 0.094) but $\hat{\beta}_1$ and $\hat{\beta}_2$ were closer, underestimating the actual by 2 times (0.392 compared to 0.554; 0.207 compared to 0.477).

### 3.2.3 Summary of Experimental Results

Our experiments show that the model's false positive and false negative rates and prior can be estimated on unknown data sets with the Hui-Walter method. Although we assumed the default distribution $Beta(1,1)$ for the errors and priors, the results from the two experiments are generally promising for rough order of magnitude estimates. With more iterations or a more specific distributional assumption, the results would improve. Results would also likely be more favorable with better trained models as well.

### 3.3 Hui-Walter Online

Our main goal in this research is to answer how we estimate false positive and false negative rates of binary categorizers to a stream of unlabeled data without available ground truth. Current online machine learning theory still assumes that labels are available for the data in question (Gomes et al., 2019). Figure 6 shows an experiment where we streamed the Wisconsin Breast Cancer Data set with the `river` python library for

online machine learning (Montiel et al., 2020) [3]. We trained a support vector machine using the mean radius and mean texture and a Random Forest Classifier using mean concavity and mean symmetry. The mean radius of the features, the mean texture, the mean radius and the mean texture were chosen, so each model had features from one leg of the dendrogram into which the features of this data set fall into (Pantazi et al., 2002). These features are different from the features used in the non-streaming experiment. The support vector machine is 90% accurate with $\alpha_1 = FPR = 0.18$ and $\beta_2 = FNR = 0.06$. The Random Forest is 88% accurate with $\alpha_2 = FPR = 0.23$ and $\beta_2 = FNR = 0.09$.

The categorizers then applied predictions to the online data and, for each time step, the contingency table was updated, the new sample was recorded, and latent profile analysis was applied to the new sample, plus the complete history of the samples. Because latent profiles are calculated when a new sample arrives, the latent class of a single sample will often flip. Due to this, Figure 6 shows the history of the parameters of interest calculated for the last latent profile. Additionally, due to the instability of the discriminant for Hui-Walter, the first 250 time steps are omitted because they contain discontinuities. The literature on Hui-Walter states that Hui-Walter may be better at estimating the test agreement than the actual false positive and false negative rates (Bertrand et al., 2005), and the results of our experiment in Figure 6 show the parameters that come close to the desired metrics of the training environment scaled up or down by a factor of 2 for $\alpha_i, \beta_i$ for $i = 1, 2$. Prior probabilities $\theta_1, \theta_2$ are more stable. Empirically, the priors are $\theta_1 = 0.11$ and $\theta_2 = 0.51$, calculated by taking the number of true values from the labels available in our experimental setup.

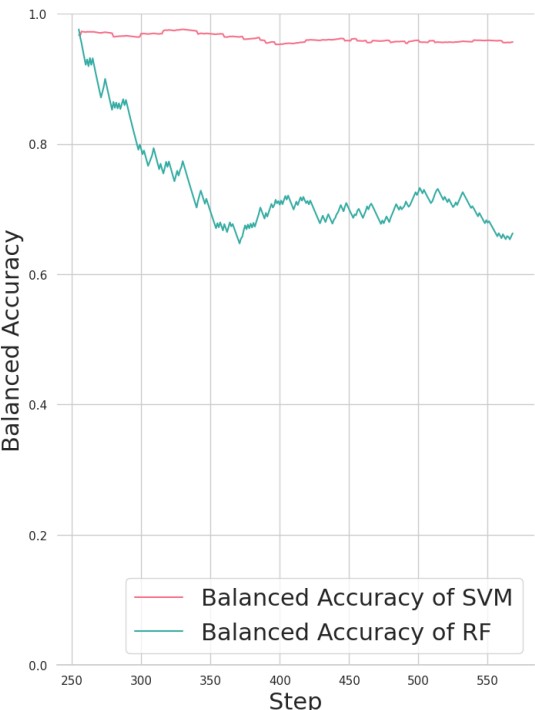

Figure 6: Online-Hui Walter applied to the Wisconsin Breast Cancer data set in a stream with the `river` python library for online machine learning Montiel et al. (2020) to simulate online data and the balanced accuracy for both classifiers was calculated over two latent subpopulations starting after a burn in period of 250 streaming samples.

Table 9 contains the mean absolute error of the values estimated with Hui-Walter versus those estimated in the training environment. We calculate the mean absolute error using the last 200 steps. This technique is similar to the burn-in period in Bayesian statistics.

---

[3]https://github.com/online-ml/river

The Rand index (RI) is a measure of the similarity between two cluster assignments, typically the ground truth labels and the clustering results Rand (1971). It is calculated by considering the number of pairs of data points that are either in the same cluster in both assignments (true positives, TP) or in different clusters in both assignments (true negatives, TN). Given the total number of pairs of data points, N, the Rand index can be computed as follows.

$$RI = \frac{TP + TN}{\binom{N}{2}}$$

The RI ranges from 0 to 1, where a value of 1 indicates perfect agreement between the two cluster assignments, and a value of 0 suggests that the assignments are completely dissimilar.

Now, let us discuss the relationship between the Rand index and accuracy. In the context of binary classification, precision is the proportion of true positive (TP) and true negative (TN) instances out of the total number of instances (P). It can be calculated as:

$$\text{Balanced Accuracy} = \frac{TPR + TNR}{2}$$

In clustering problems, the Rand index can be considered a measure of the "accuracy" of a clustering algorithm when compared to the ground truth labels. However, it is essential to note that there are some differences between these two measures.

The Rand index is designed for clustering problems and considers both true positive and true negative pairs, whereas the accuracy is designed for classification problems and is based on the number of correctly classified instances. The Rand index is a measure of similarity between two cluster assignments and does not directly evaluate the quality of a single clustering result. On the contrary, accuracy directly evaluates the performance of a classification model. In summary, while the Rand index shares some similarities with accuracy, they are designed for different types of problem (clustering versus classification), and their interpretations and use cases are not entirely the same.

| Classifier | Balanced Accuracy | Rand Index | Accuracy |
|---|---|---|---|
| RF | 0.96 | 0.78 | 0.86 |
| SVM | 0.73 | 0.82 | 0.90 |

Table 7: Wisconsin Breast Cancer Data Set: Comparison of the mean Balanced Accuracy estimated by Hui-Walter on the online data, Rand Index, and Ground Truth Accuracy for Random Forest and Support Vector Machine

| Classifier | Balanced Accuracy | Rand Index | Accuracy |
|---|---|---|---|
| LR | 0.76 | 0.71 | 0.82 |
| RF | 0.77 | 0.70 | 0.81 |

Table 8: Adult Data Set: Comparison of the mean Balanced Accuracy estimated by Hui-Walter on the online data, Rand Index, and Ground Truth Accuracy for Logistic Regression and Random Forest

In order to compare our method to established methodologies from unsupervised learning, we compare the balanced accuracy found for our online implementation of Hui-Walter with the Rand Index which has been shown to relate the accuracy to the unsupervised learning setting (Garg & Kalai, 2018). For the Wisconsin Breast Cancer data set, we compare the balanced accuracy for the support vector machine and the random forest to the Rand index for both classifiers over both latent sub-populations. For the random forest model, we found a Rand index of 0.78 and a balanced accuracy of 0.96 from Hui-Walter, and the ground truth accuracy is 0.86. For the support vector machine, the balanced accuracy is 0.73, the RI is 0.82, and the ground truth accuracy is 0.90. These can be seen in Table 7 where we can see that online Hui-Walter gives a clear improvement over perhaps the only available baseline metric for this problem. The results are even more

compelling for the analogous experiment run on the adult data set; as we can see in Table 8, the Balanced Accuracy is closer to the ground truth accuracy than the Rand index. The classifiers in question (Random Forest, Support Vector Machine, and Logistic Regression) were all chosen for their simplicity and ubiquity in production settings.

Table 9: Mean Absolute Error for Streaming Parameters

| Parameter | MAE |
|-----------|------|
| $\theta_1$ | 0.35 |
| $\theta_2$ | 0.03 |
| $\alpha_1$ | 0.10 |
| $\alpha_2$ | 0.15 |
| $\beta_1$ | 0.05 |
| $\beta_2$ | 0.09 |

Although this test yielded reasonable estimates for most parameters, the main issue with this approach for online estimation is the need for more standard errors, which is given by Bayesian estimation for the static case (Johnson et al., 2001). The worst estimate on the streaming data, $\theta_1$, also closely matches the static data case found with the Gibbs sampling method.

## Conclusion

Our results show that the Hui-Walter method works very well for static data and gives *plausible* results for online data, and this method requires further improvements. The data from this experiment are in a three-way contingency table, and a log-linear model could improve the estimate as is in (Hui & Zhou, 1998). Additionally, the algebraic geometry of the three-way contingency tables could yield different results, as there is a relationship between tensor factorization and the parameters of the product-multinomial model (Dunson & Xing, 2009). One of the limitations of using explicit formulas for the $2 \times 2 \times 2$ case is that sometimes the solutions do not have *plausible* solutions (Hui & Walter, 1980). In other words, it is possible to get a solution for the false positive rate greater than one or less than zero. Furthermore, the performance of the straightforward solutions on the online data lacked confidence intervals, which are problematic for applied settings, and the solutions are not as precise as the Gibbs sampler. One suggestion for further research on how this can be improved is to leverage online Gibbs sampling (Kim et al., 2016; Dupuy et al., 2017).

We have shown how to use the Hui-Walter paradigm to estimate false positive and false negative rates and prior probabilities when no ground truth is available. One of the core assumptions of statistical reasoning and its offshoot machine learning is that all of the models' possible data is collected ahead of time, and a practitioner only needs to sample appropriate training and holdout test sets. This assumption is far from the reality of many machine learning applications worldwide. Machine learning practitioners increasingly adapt to the reality that models make predictions based on previously unseen unlabeled data that change in the shape of the distribution over time. In this paper, we tackle this very common problem in applied machine learning. Specifically, we derived a way to measure the efficacy or effectiveness of machine learning models in a situation where the data set is unknown and there are no assumptions about the data distribution. We have also shown how this methodology applies to static and streaming data.

### Broader Impact Statement

This research is expected to significantly impact the field of applied machine learning by providing a methodology to assess the efficacy of the model in the common but challenging circumstances of incomplete, unknown, or dynamically changing data sets. The Hui-Walter method, despite its limitations, as highlighted in our study, offers a robust starting point for practitioners dealing with data ambiguity. Additionally, our explorations of potential improvements, such as the application of online Gibbs sampling, open a promising path towards refining this technique. Ultimately, the ability to gauge the effectiveness of machine learning

models even in the absence of complete data will empower practitioners across industries, driving more confident and informed decision-making processes.

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
