# OpenReview forum: "Unlocking Unlabeled Data: Ensemble Learning with the Hui-Walter Paradigm for Performance Estimation in Online and Static Settings"
_TMLR — Rejected by TMLR_

### Review · Reviewer_b9BA · 2023-07-21

**Summary Of Contributions:**

In this work authors adapt the Hui-Walter paradigm which enables to estimate key performance metrics such as false
positive rate, false negative rate, and prior with unlabelled data extending this paradigm for handling online data, opening up new possibilities for dynamic data environments.

**Audience:**

Yes

**Broader Impact Concerns:**

The current section included by the authors is ok.

**Claims And Evidence:**

Yes

**Requested Changes:**

I think that the experimental part should be improved both in therms of quantity of experiments that comments.

**Strengths And Weaknesses:**

The first part of the paper, even if not particularly novel, is well written and presented and interesting for practitioners.
Nevertheless, the second part, is a bit week: just two datasets which are old and simple. A deeper experimental results section I think is needed to improve the strength of the paper.

---

### Review · Reviewer_2obe · 2023-07-29

**Summary Of Contributions:**

The paper proposes a novel approach to estimate key performance metrics in scenarios where no ground truth is available, utilizing the Hui-Walter paradigm. The authors demonstrate the effectiveness of the proposed approach through extensive experimentation on both static and online datasets. The primary contributions of this research lie in its successful application of the Hui-Walter paradigm to the field of machine learning.

**Audience:**

Yes

**Broader Impact Concerns:**

None.

**Claims And Evidence:**

No

**Requested Changes:**

(1) Provide a comprehensive explanation of the Hui-Walter paradigm.

(2) Incorporate a comparative analysis with existing approaches.

(3) Elaborate on crucial aspects while eliminating redundancy.

**Strengths And Weaknesses:**

Strengths:

(1) The paper proposes a novel approach to estimate key performance metrics in scenarios where no ground truth is available, using the Hui-Walter paradigm. This approach makes an impact on the field of applied machine learning by offering a methodology to assess model efficacy under challenging circumstances, where ground truth data is unavailable or dynamically changing.

(2) The paper presents experimental results on both static and online datasets, showcasing the effectiveness of the proposed approach. The results indicate that the Hui-Walter method performs exceptionally well on static data and yields plausible results for online data.


Weaknesses:

(1) The paper focuses on adapting the Hui-Walter paradigm to the field of machine learning, limiting its scope to binary classification problems and potentially restricting its direct applicability to multi-classification problems.

(2) The paper lacks a comparison of the proposed approach with existing methods, hindering the assessment of its effectiveness.

(3) The paper's writing logic is confusing, containing redundant content that can cause confusion for the reader. The organization of the paper should be adjusted. For instance, Section 2 consists of only a paragraph with three sentences, and there are repeated subtitles in subsections 4.1 and 4.3.

---

> ### Comment · Reviewer_2obe · 2023-09-25
> **Response to the revision**
>
> Thanks for the authors' response. However, in the revision, the authors do not fully address my concerns. The writing of this article is redundant and fails to highlight the necessary key information. Effective comparison algorithms are lacking to verify the effectiveness of this method. While the idea in this manuscript is interesting, it is limited to binary classification problems.

---

### Review · Reviewer_jBMD · 2023-08-29

**Summary Of Contributions:**

The paper proposes a technique to estimate false positive rate (and false negative rate) of a set of binary classifiers using only unlabeled data. The methodology can handle data available all at once or coming in an online fashion. Numerical experiments indicate that the methods offer some benefits.

**Audience:**

Yes

**Broader Impact Concerns:**

There are no societal impact concerns.

**Claims And Evidence:**

No

**Requested Changes:**

The primary changes I recommend are:
- Reading Section 3.2's description of maximum likelihood estimation (MLE), I have a difficult time distinguishing what is literature review from what is proposed methodology. Perhaps you can separate into a "review" part and "proposed" part.

- The equations at the end of page 4 seem to be the heart of the methodology, but they are not distinguished from the rest of the paper. It would be good to streamline Section 3.2 so that these equations are mentioned as soon as possible, and assign them some labels (in the sense of label{eq:..}) which makes it easy to refer to them later on.

- Content such as the equation for the multinomial probability mass function (page 4) is standard knowledge, and do not need to be mentioned in displayed math mode.

- Based on the last paragraph of Section 3.2, it seems that the theoretical guarantees that come with the Hui-Walker framework are predicated on some assumptions: it would be good to bring this to the forefront upon first mentioning of the Hui-Walker paradigm.
  - more generally, it would be good to be very clear on a) the number of sub-populations necessary and b) number of classifiers necessary and c) assumptions on the different classifiers.

- I am not clear on why Gibbs sampling is necessary for maximum likelihood estimation. I have seen Gibbs sampling being used to approximately draw from the probability distribution of interest, such as the Bayesian posterior. In this case, I do not see a probability distribution of interest.

- I recommend labeling the most important equations (something like a begin{equation} label{eq:eq1} end{equation} should do the job). Later on in the paper, you can use ref{eq:eq1} to link to the important equations.

 - The second section (our contributions) is very short, and does not meaningfully add information found elsewhere in the paper. It would be better to remove this short section.

- Remove the nearly repeated sentences. A non-exhaustive list of them include
  - The second, third and fourth paragraphs of Section 4.1.1 have much overlap in describing the importance of the Wisconsin breast cancer dataset.
  - The top of page 4 says "Maximum likelihood estimation (MLE) is a method for estimating the parameters of a statistical model given a set of observations". The end of page 5 nearly repeats this "Maximum likelihood estimation is a method used to estimate the parameters of a statistical model given a set of observations. In the case of product-multinomial distributed data, this involves finding the values of the parameters that maximize the likelihood of observing the provided data".

- Please remove the link to the github repo: we are expecting a double-blind submission, but the repo breaks the anonymity on the authors' end.

Here is the list of secondary recommendations.
- There are some broken table/figure references
  - In Section 4.3.1, "Underneath Table 4" should probably be fixed as "Underneath Table 1".

- Improve the legibility of the figures
  - font sizes in figures (label sizes, legend) are too small compared to the font size of the main text
  - there is a lot of white space right after the caption of Figure 5. this white space should be removed.

- Smooth out the transition between different sentences
  - For example, "Using the above formulas, we can increment the streams’ counts and apply the estimates to streaming data, allowing us to examine the false positive and false negative rates, and prior probabilities of streaming data on a given time step. Boosting is a family of algorithms that convert multiple weak learners in parallel to strong learners. With Hui-Walter online, we can leverage multiple weak learners to assess the false positive and false negative rates, and prior probabilities to improve the predictions by making them "prior aware" of the unlabeled data streams." The second sentence does not follow from the first sentence, and the third does not follow from the second.

**Strengths And Weaknesses:**

# Strengths
The problem of evaluating the performance of a predictive system is important in both research and production. The setting of unlabeled data is also prevalent. Hence, the topic is highly relevant.

# Weaknesses
I think the "proposed method" section could be strengthened so that by the end of reading, a reader is very clear about the steps involved in your estimation process. Currently, it is not clear.

The "experimental results" section is not clearly communicating what a reader should learn from looking at the attached tables/figures. For example, I could not find associated captions or discussions for Table 1. I would like the authors spell out more clearly whether the given set of "Experimental Results" is a good enough approximation of "True Values": there are around 10 numbers being given in a tabular form, and I am not sure what to make of them.

The paper has some nearly duplicate sentences, particularly in the experiments. For instance, the top of page 4 says "Maximum likelihood estimation (MLE) is a method for estimating the parameters of a statistical model given a set of observations". The end of page 5 nearly repeats this "Maximum likelihood estimation is a method used to estimate the parameters of a statistical model given a set of observations. In the case of product-multinomial distributed data, this involves finding the values of the parameters that maximize the likelihood of observing the provided data".

---

> ### Comment · Reviewer_jBMD · 2023-09-15
> **Acknowledgment of Rebuttal**
>
> I thank the authors for responding to my suggestions. However, my overall assessment is that the paper is not yet ready for publication.
>
> - Points of improvement compared to the previous draft include
>   - more interpretation of experimental data given in the form of tables
>   - removal of duplicate exposition
>   - more legible figures
> - I would like to see the next draft make the following changes, among others:
>   - explain how the methodology can be applied the multiclass classification setting, and provide some empirical evidence
>   - distinguish between previous work and proposed methodology:
>     - in Section 2.2.1, Hui-Walter as written is not an innovation compared to the existing techniques. It appears that only section 2.2.2 is a novel methodology

---

### Decision · Action_Editor_jZoV · 2023-10-21

**Recommendation:** Reject

**Comment:**

After considering the revision and the author's rebuttal, the reviewers judged that the paper still required a major revision to be made acceptable to TMLR (see their comments). The paper should be significantly rewritten taking the reviewers' feedback in consideration, including also a more complete empirical section. The reviewers also noted that the writing was too redundant: the current length of the paper (16 pages) is not justified.

**Audience:**

Due to lack of clarity of the writing, the interest is fairly limited. The reviewers also suggested to discuss how the author's approach could be adapted to multiclass classification, in order to increase the applicability and interest in the work.

**Claims And Evidence:**

Not enough. The reviewers raised concerns that the writing was not clear enough and this obfuscated the claims and its evidence. One reviewer also noted that the two proposed datasets used to test their method were too simple to sufficiently demonstrate its effectiveness.

**Resubmission Of Major Revision:**

The authors may consider submitting a major revision at a later time.